# Sports and the Pandemic: The Impact of COVID-19 on Active Living and Life Satisfaction of Climbers

**DOI:** 10.3390/ijerph20031964

**Published:** 2023-01-20

**Authors:** David Jungwirth, Daniela Haluza

**Affiliations:** Department of Environmental Health, Center for Public Health, Medical University of Vienna, 1090 Vienna, Austria

**Keywords:** Austria, Germany, COVID-19 crisis, physical activity, health, nature, sitting time, exercise

## Abstract

The COVID-19 pandemic has resulted in significant changes in every aspect of our lives. Because of the measures imposed, people were only allowed to leave their homes for certain purposes, and all types of cultural and sports events were canceled. Climbers were greatly affected by these limited options for regular physical activity outside of the home environment. Little is known about the crisis’ effects on the climbing community in German-speaking regions. Thus, we surveyed 1028 German-speaking climbers (mean age 34.6 years, SD 10.4; 50.4% females) from December 2020 to February 2021. A cross-sectional online survey collected data on climbing frequency and preferences as well as levels of life satisfaction, using the standardized Short Life Satisfaction Questionnaire for Lockdowns (SLSQL) before and during the crisis. Results showed that due to the pandemic, study subjects climbed less frequently, preferred outdoor locations to climb, and showed decreased life satisfaction scores (21%, (d = 0.87, *p* < 0.001). In conclusion, these findings highlighted that the COVID-19 pandemic had a negative effect on climbing sports activities and life satisfaction in this study sample. To preserve physical and mental health, indoor and outdoor sport activities should be continued as much as possible with reasonable hygiene concepts in place.

## 1. Introduction

In March 2020, the COVID-19 pandemic caused massive restrictions on everyday social life as we knew it [1]. People all over the world were obliged to reduce activities in public spaces to a minimum or even stay at home to avoid human contact. Due to stay-at-home orders, lockdowns, and social distancing measures, indoor as well as outdoor sport facilities such as fitness centers, sport clubs, climbing gyms, and golf courses were closed for many weeks or even months [2]. Recently, many international studies reported on the negative impact of COVID-19 on the quantity of physical activity, while screen-related sedentariness increased [3,4,5]. There is also scientific evidence that life satisfaction, quality of life, mental health, and well-being decreased due to COVID-19 [4]. Life satisfaction is closely related to happiness and a crucial part of subjective well-being assessed in terms of current mood, achieved goals, self-concepts, and self-perceived ability to cope with life [6,7,8]. This construct is a suitable measure of psychosocial strain during the situational context of curfews and lockdowns. Studies on the effects of the COVID-19 pandemic consistently report decreased life satisfaction scores worldwide [6,7,9,10]. Potential reasons include decreased social interactions with relatives and acquaintances, mobility restrictions, and closed sports and wellness facilities [6,8,11,12,13]. Moreover, it is commonly accepted that reduced levels of physical activity results in reduced life satisfaction [9].

The closure of indoor fitness studios led to a measurable change of physical activity patterns towards outdoor sports and home-based training [14]. Health awareness was a relevant factor driving indoor home exercise during the COVID-19 pandemic [15]. Although physical activity during lockdowns was important for health and wellbeing, training at home differs from the usual setup, e.g., in a gym, and might be more challenging for people who have to be considerate of family members, flat mates, or pets. Additionally, it is not possible to practice all sports in the home environment, for example, installing a climbing wall at home or playing golf [16]. 

It is well known that regular physical activity is essential for health promotion and the prevention of non-communicable disease in all age groups [17]. For sustainable physical and mental health benefits, international fitness guidelines recommend at least 150 min/week of moderate-to-vigorous-intensity physical activity [18,19]. The main goal of this article was to study the impact of COVID-19 on active living and life satisfaction on people that are regularly physically active with the example of climbing. Wheatley and Bickerton rated climbing as a moderately intensive sport [20]. Climbing, either practiced indoors or outdoors, on artificial walls or natural surfaces, is thus useful for improving physical fitness parameters [21]. Various physiological changes include increased mobility, strength gains, and better muscle balance, and climbing is thus increasingly used in the context of sports therapy, e.g., in spinal column patients and mental illnesses [22]. Climbing requires both muscle strength and problem-solving skills. Typical body type characteristics observed in professional climbers determine a favorable strength-to-weight ratio such as low skinfold thickness and low body fat [23]. Well-trained forearm flexors with a high aerobic capacity are particularly important for efficient climbing. 

Many people kept in shape during the lockdown period with walking, running, and fitness exercises, with or without the support of digital offers [10]. Impacts of the pandemic on general recreational sports patterns and social interactions have been studied in different contexts all over the world [6,16]. However, the climbing community is currently understudied, despite the associated economic investments and job opportunities in recent decades [24]. Indoor climbers were also affected by the closures of fitness and sports facilities in March 2020. As an example, a popular climbing gym in Vienna was closed until 2 June 2020, and then reopened with very restrictive COVID-19 mitigation measures for athletes including keeping a one-meter distance in the gym, keeping a two-meter distance when climbing, hand disinfection, only climbing with personal equipment, and closure of showers [25]. These measures were important gor preventing virus transmission in the climbing gyms. 

So far, knowledge on the influence of the pandemic on climbing, a popular mass sport with different indoor- and outdoor-based styles, is scarce. Based on the pertinent literature on the negative effects of the crisis on physical activity and mental health, we hypothesized that adult climbers in German-speaking countries were also affected by lockdowns, curfews, and gym closures. We were also interested in elucidating whether socio-demographic variables such as gender and age impacted perceived impacts of the pandemic on health, climbing, and life satisfaction levels. Thus, the current study examines whether climbing frequency and preferences of indoor or outdoor climbing facilities as well as levels of life satisfaction changed during the COVID-19 pandemic. Using a cross-sectional online survey including the standardized Short Life Satisfaction Questionnaire for Lockdowns (SLSQL), we surveyed a nonrepresentative convenience sample of German-speaking climbers and compared ratings before vs. during the COVID-19 crisis in the climbing context [6,10,13].

## 2. Methods

### 2.1. Study Design

This non-representative online study among a cross-section of German-speaking adult climbers assessed self-reported prevailing perceptions regarding physical activity and life satisfaction in the context of the COVID-19 crisis. The survey was designed using the Checklist for Reporting Results of Internet E-Surveys (CHERRIES) [26]. Study participation was voluntary. No incentives were offered. Randomized or adaptive items were not used. Prior to data collection, ethical approval was granted from the institutional ethical committee of the Medical University of Vienna, Austria, on 9 October 2020. The study was conducted following the ethical standards laid down in the Declaration of Helsinki.

To review the completeness and comprehensibility of the survey, 15 voluntary participants, climbers as well as non-climbers, pretested the online survey. The adapted online survey was freely accessible via the web-based survey tool SoSci Survey from 5 December 2020 to 12 February 2021 [27]. The online survey in German was started by 1320 participants, and 1084 fully completed the survey (82.12% completion rate). Of these, 56 data sets were excluded from further analysis, as these participants indicated that they did not climb before the COVID-19 pandemic, and the survey ended for these non-climbers. The average completion time of the survey was 7.39 min (SD 2.26).

The first page informed survey participants about the study aim. All study subjects indicated their informed consent before starting the online survey. An explanatory text defined that “before COVID-19” means the time period before 16 March 2020, and “during COVID-19” means the time period after this date [4]. The survey items were adapted to climbing from previously used questionnaires on physical activity and in view of COVID-19-related recommendations for safe climbing [10,14,16]. The link to the online survey was distributed using the snowball system via commonly used social media pages and designated groups on WhatsApp, Facebook, and Signal addressing climbers and outdoor enthusiasts. The website administrators of the most prominent climbing organizations, such as the Austrian Alpine Club, operating more than 200 climbing facilities throughout Austria [28], Nature Friends [29], the Austrian Climbing Association [30], and the Vienna Climbing Hall [25], were contacted to send the questionnaire link via their official newsletter and social media platforms.

### 2.2. Measures

A priori, the dichotomous filter question “Did you regularly climb before the COVID-19 pandemic?” (answer options: yes or no) distinguished between climbers and non-climbers. For those who selected “no”, data collection stopped at this point. For the others, the first part of the online survey collected socio-demographic characteristics (single choice) such as gender (male, female, or non-binary), age (in years), highest education level (primary, secondary, or tertiary), country of residence (Austria, or others by using a free text box), and area of residence (rural area, town, or city). 

Four single items used a 5-point Likert scale ranging from 1 (strongly disagree) to 5 (strongly agree) to collect the degree of agreement with the following statements: (1) “I think the COVID-19 pandemic is dangerous”, (2) “Due to the COVID-19 pandemic, I am worried about my health”, (3) “Due to the COVID-19 pandemic, I’m worried about getting infected while climbing”, and (4) “Due to the COVID-19 pandemic, I disinfect my hands before and after climbing”. 

Further, we asked for activities both before and during the COVID-19 pandemic (single choice) regarding these three categories: (1) predominant climbing venue (climbing gym, nature, others, or not at all), (2) predominant climbing style (bouldering, sport climbing, or alpine climbing and others), and (3) predominant outdoor sports frequency (0–1 h, >1–2 h, >2–5 h, or >5 h). 

The last part of the questionnaire assessed life satisfaction of the study subjects before and during the COVID-19 pandemic using the standardized Short Life Satisfaction Questionnaire for Lockdowns (SLSQL) [6,10,13]. The response options of this crisis-oriented questionnaire ranged from 1 (strongly disagree) to 5 (strongly agree). The SLSQL consisted of three items: (1) “In most ways, my life is close to my ideal”, (2) “So far, I have gotten the important things I want in life”, and (3) “I am satisfied with my life”. Participants rated their agreement with these statements both before and during the pandemic to allow for comparison.

### 2.3. Statistical Data Analysis

Descriptive statistics were used to report categorical data as absolute and relative frequencies and continuous data as mean and standard deviation (SD). All statistical analyses were performed using the statistical software SPSS Statistics for Windows, Version 27.0 (IBM Corp., Armonk, NY, USA). Statistical significance was set at *p* < 0.05. Independent-samples T tests were used to compare ratings between the gender and age subgroups. Paired-samples T tests were used to assess the magnitude of the changes in responses between the before-COVID-19 and during-COVID-19 periods. To evaluate the changes in preferences for climbing venue, climbing style, and outdoor spot frequencies as a result of COVID-19, Wilcoxon signed-rank tests were conducted. e-Effect sizes were calculated using Cohen’s d to define the magnitude of the change score, and internal consistency was measured using Cronbach’s alpha. Both measures were interpreted as small (i.e., 0.2), moderate (i.e., 0.5), or large (i.e., 0.8). Internal consistency (Cronbach’s alpha) for the three SLSQL items before COVID-19 (i.e., 0.841) and for the three SLSQL items during COVID-19 (i.e., 0.889) were high.

## 3. Results

### 3.1. Socio-Demographic Characteristics of the Sample

The final sample included 1028 participants, with an almost equal split between females (n = 518, 50.39%) and males (n = 507, 49.32%), and three non-binary people (0.29%, Table 1). The average age of participants was 34.56 years (SD 10.42, Md 32 years, ranging from 18 to 72 years). The median split at 32 years yielded two age groups, i.e., Age low (n = 529, 51.5%) and Age high (n = 499, 48.5%). Most study subjects lived in Austria, with the rest in other German-speaking countries, and the majority had a tertiary education level and lived in rural areas. 

### 3.2. Perceived Effects of COVID-19 on Health 

Table 2 shows subgroup-specific responses (1: strongly disagree to 5: strongly agree) to single items related to the perceived effects of the COVID-19 pandemic on climbers. Notably, all effect sizes were negligible or small. In total, average scores of 3.82 (SD 1.14) were found, with 65.95% of participants stating to agree/strongly agree that the COVID-19 pandemic is dangerous, with no statistically significant gender or age group differences. More than half of the participants (50.19%) indicated to strongly disagree/disagree about being worried about their health, with statistically significant higher average ratings for females compared to males (*p* < 0.001). Further, 73.54% strongly disagreed/disagreed about being worried about getting infected with COVID-19 while climbing, with higher ratings in females compared to males and younger participants compared to older ones. Moreover, 49.03% of the study subjects agreed/strongly agreed about disinfecting their hands before and after climbing as a response to the COVID-19 pandemic, with statistically significant higher average ratings among females compared to males (*p* = 0.008). 

### 3.3. Perceived Effects of COVID-19 on Climbing 

Table 3 depicts responses to items related to climbing (single choice) before and during the pandemic. Due to COVID-19 restrictions, the participants replaced climbing gym visits (−33.50%) largely by climbing in nature and non-climbing (22.60 and 10.40%, respectively). While reported frequencies of bouldering and sport climbing decreased, alpine climbing and other climbing styles increased in popularity (12.60%). Total time spent outdoors for physical activity decreased in favor of an increase in shorter time activity (8.20% increase). Wilcoxon signed-rank tests revealed a statistically significant change in preference for climbing venue (z = −18.099), climbing style (z = −9.116), and outdoor sports (z = −7.243) in the before and during COVID-19 comparison, and this was also true for gender and age subgroups (all: *p* < 0.001).

### 3.4. Life Satisfaction Change in Climbers

Table 4 summarizes the responses to the SLSQL, comparing scores before and during the COVID-19 periods. Paired-samples *t* tests showed significant decreases for all items when comparing before and during COVID-19 scores, with all *p* values smaller than 0.001. There was a statistically significant decrease in the total score of Life Satisfaction by 20.97%, with a large effect size during compared to before COVID-19 (t = 72.90, d = 0.87, *p* < 0.001). We reported similar observations for the gender and age groups (all: *p* < 0.001), with females showing the highest decrease in Life Satisfaction (−22.82%).

As for between-groups comparisons, the independent-samples T test did not reveal statistically significant gender differences in Life Satisfaction before COVID-19, but during COVID-19, with lower average scores for females (mean 3.26, SD 1.32) compared to males (mean 3.46, SD 1.25, *p* = 0.013). There was no statistically significant difference for age groups in their ratings for both time points (all: *p* > 0.05).

## 4. Discussion

The present study examined the effects of the COVID-19 pandemic on active living and life satisfaction in German-speaking climbers. This unprecedented health crisis fundamentally changed everyday life and impacted people’s former patterns of recreational activities, eating behavior, and social interactions [13]. As a result, measures of health and wellbeing decreased substantially in response to the crisis [5,31]. In our study, the participants climbed significantly less frequently, climbed predominantly in natural environments, and conducted less outdoor sport in general during the COVID-19 pandemic than before. Likewise, a study by García-Tascón et al. showed decreased quantity and intensity of physical activity before and during COVID-19 restrictions [31]. Schnitzer et al. showed that the Tyrolean population were less physically active in general and preferred to go for walks during the curfew [14]. The findings of these studies on the general population are in line with the present study that highlights an overall decrease in physical activity among climbers.

Reflecting the longer-term closures of indoor fitness facilities, previous research found a predominant trend towards home-based exercising [10,13,32]. These results are in disagreement with the findings presented here in our sample of climbers. Other related studies found increased visits to urban green spaces, urban forests, and rural areas for physical activity in the general population [10,33]. A study by Rice et al. found that outdoor exercising by US climbers decreased during the COVID-19 pandemic, with social distancing recommendations as the main reason for the activity changes [34]. Contrarily, this study showed that participants mainly climbed outdoors during the pandemic when compared to before. This observation suggests that the COVID-19 pandemic encouraged climbers to switch to outdoor routines as a feasible alternative to indoor climbing facilities, which were closed or had restricted access due to COVID-19 regulations such as closure of showers. This is presumably rooted in the difficulty of practicing climbing at home due to spatial and structural limitations in an average home. So, it is more likely that climbers preferred to climb outdoors than to purchase and install climbing equipment for domestic use, also mirroring the wish for meeting former climbing mates and social interaction while climbing [34].

Individual preferences vary in relation to gender, age, local climbing trends, and availability of near-home climbing gyms or natural climbing areas. Nevertheless, climbers slightly prefer climbing indoors (55% vs. 45%) to climbing outdoors, with females favoring indoor climbing, while males reported to participate more in outdoor bouldering [35]. Outdoor climbing allows people to spend time in nature. Compared to indoor climbing, outdoor climbing offers different external conditions, such as weather and temperature, as well as the opportunity to stay in natural environments and share nature with other living beings. Under different environmental conditions, human physiology and psychology adapt and react to the surrounding environment. Many studies show that being in nature has a positive effect on physiological stress [36]. These mental aspects are important for athletes as climbing is a sport that requires high levels of mindfulness, concentration, and stress management capacities.

Climbing is especially popular among younger adults, an age group that is not the major risk group of COVID-19-related morbidity and mortality. However, transmission risks to vulnerable people as well as the unprecedented risk for the post-COVID-19 syndrome even in the younger and healthy part of the population exist. Although two-thirds of the surveyed climbers agreed that the COVID-19 pandemic was dangerous, the scores for being worried about one’s health and getting infected while climbing where much lower. The reasons could be twofold: first, the subjects shifted to climbing outdoors and felt adequately protected by the conditions offered by a natural environment, and second, climbing can be easily done without very close human contact. These considerations suggest that climbing was perceived as relatively safe in terms of COVID-19 transmission and infection, given that members of the climbing community are normally quite young and healthy. This is also supported by the finding that less than half of the participants agreed to disinfect their hands before and after climbing. However, this observation shows that there is still room for improvement in this respect.

Indoor climbing requires frequent touching and holding of the artificial climbing surfaces. In their microbiological analysis of climbing wall holds, Bräuer et al. detected microorganisms originating from soil, but also human skin on all investigated surfaces [37]. Especially for indoor climbing, some athletes might be concerned about the standard of protection regarding a COVID-19 infection, which has to be ensured by the gym operator, but requires a high commitment and adherence by the athletes as well. Liquid chalk, which is a type of climbing chalk that is made with a mixture of powdered chalk and alcohol, has become increasingly popular among climbers. The alcohol in liquid chalk is used as a carrier to help keep the chalk on the hands and can be found in varying concentrations (from 40–80%). It is important to note that liquid chalk is primarily used to dry and grip the hands, not to decrease the risk of infection. Consequently, hygiene recommendations in times of a virus pandemic should include disinfecting and washing hands before and after climbing, and cleaning surfaces such as climbing holds frequently and regularly [38].

As for gender differences, Castañeda-Babarro et al. found that men showed a greater decrease in physical activity than women during the COVID-19 restrictions in Spain [32]. In line with these findings, Romero-Blanco et al. demonstrated that female students spent significantly more time in physical activity during the lockdown compared to male students [39]. In contrast, the present study showed that females reported lower climbing frequencies than males during the COVID-19 pandemic. These gender effects might root in different motives for physical activity, with elements such as competition or social recognition being a major driver in male activity patterns. This finding could speculatively explain in part the higher decrease in life satisfaction in females when compared to their male counterparts.

As already mentioned, as a worldwide phenomenon, the intensity and frequency of physical activity decreased during the periods of confinement [3,4,5,9,12,13,31]. On the other hand, physical activity during the crisis had a protective effect on perceived psychological distress associated with the pandemic [40]. Governments widely loosened the very strict restrictions of the first lockdown, also including the closure of public parks, step by step in accordance with public health experts, who brought into play the known benefits for physical and mental well-being of being outdoors in nature [11,36,41,42]. Interactions with natural environments in addition to being physically active exert a comforting and relaxing effect, reduce anger, fear, and stress, and increase pleasant feelings. Studies have already attested the health-related relevance of high-quality urban green spaces as cost-free and valuable alternatives to indoor and outdoor sports venues in times of the crisis [10,11,12,33]. Acknowledging the positive health effects of regular recreational exercising with the example of climbing, this study, conducted in 2020/2021, adds much needed knowledge on perceptions and behavioral changes to the rapidly increasing number of scientific reports on the physio-psychosocial effects of the pandemic [8,11,12].

Due to the curfews, a range of arts, cultural, and sports events such as exhibitions, music festivals, and football games were canceled without replacement. The engagement in these activities and access to cultural goods or at least the option to participate in them, as well as the symbolic value of their existence, fosters life satisfaction [8]. Although it is assumed that people tend to adapt to the new situations and norm, and change their behaviors, expectations, and happiness set points, this requires some time [6,9]. The impact of COVID-19 on the participation and engagement in a range of cultural events might be part of the explanation for significantly reduced scores in life satisfaction (by 21%) in the study participants.

We assessed the change in life satisfaction levels before vs. during the pandemic in the general sample and between subgroups, acknowledging that, among other factors, age and gender influence this score [7,8,9]. This is relevant as the study participants were potentially more physically active and younger, and had qualitatively greater social networks with their peers than the rest of the population. Further studies should use qualitative and mixed methods to increase the understanding of the effects of engagement with sport activities such as climbing on health and wellbeing. Nevertheless, this study contributes to elucidate the impact of COVID-19-related closures of sport venues with reference to a large sample of German-speaking climbers.

Notably, it is still too early to gauge the long-lasting effects of COVID-19 on the climbing gym industry—from supplies to memberships and closures. Nevertheless, according to the *Climbing Business Journal*, 2021 was the year with the most new climbing gym openings ever in the United States and Canada, with an increase in new costumers, also mirrored by climbing shoes and harnesses sales [24]. The pandemic did not flatten the recent years’ trend of robust climbing gym growth which has surpassed 5% every year for the past decade, but rather increased climbing’s popularity. The climbing gym industry shows a remarkable resilience in the face of the pandemic, with only 3% of climbing gyms compared with about 20 to 25% of general fitness studios having permanently closed in North America in 2020 [24]. Although single gyms did not manage to open again, the general political ambition to ensure the reopening of a broad range of fitness facilities for all strata of the population is of public health importance and in accordance with the health in all policies paradigm [43]. Austrian and German governments, for example, compensated the fitness sector with about 80% of the revenue lost during the curfews. As this financial compensation was often reinvested, athletes now profit from renovated gyms and new equipment in these countries [44].

Exercise in general is not only linked to physical, but also to mental health benefits [17,45]. Rising endorphin levels provide one reasonable neurobiological pathway for the beneficial psychological effects of workout [4]. In this regard, sport climbing might have additional positive effects as it aims at achieving the very real goal of reaching the top in a relatively short time. The associated success experience potentially causes strong emotions of having accomplished a challenge, increases self-esteem and self-efficacy, while it might also provoke respect and admiration by others. The latter is in line with the theory proposed by Buechter and co-workers that the social contact with like-minded people and peers might also play an significant role for psychological well-being effects of climbing [46]. This often leads to the development of a sense of belonging with a group and longer-lasting friendships in the climbing community. Additionally, for safety reasons, sport climbing is teamwork involving at least two people—the climber and the belayer—who are equally involved and take turns, assuming a constant amount of trust and confidence between these athletes. These social aspects are directly connected with the establishment of climbing-associated clubs and national associations. The constant rise in climbing gym openings (mainly focused on bouldering) worldwide as a correlative of the supply/demand ratio, especially driven by the popularity of this leisure activity among young, fit people [24]. The trend is unbroken, as eventually, the pandemic has highlighted the relevance of these social aspects, which is pivotal in choices of activities and time investment. This community effect is likely to be greater in smaller, more homely gyms. However, the long-term developments in the fitness sector are still unclear, with the majority of the newly opened gyms being already planned before the crisis.

## 5. Limitations

This study’s results should be interpreted in view of some methodological limitations. Given the constantly changing national and regional restrictions, we compared the perceptions of climbers in the time before the start of the COVID-19 crisis in March 2020 to the time thereafter. So, this study did not focus on distinct periods of closing and re-opening. Notably, Austria, similar to other neighboring countries, was in the third extended lockdown between 26 December 2020 and 7 February 2021. We collected survey data between 5 December 2020 and 12 February 2021. There is increasing evidence that all levels of physical activity decreased and sedentary behavior increased due to COVID-19 curfews, which was also shown in this study sample [3,4,5,9,12,13]. This assessment mainly focused on the shift of modes of activity from indoor to outdoor environments in climbers, not the physical activity level and intensity itself.

This study was designed as a cross-sectional online study using a German questionnaire. So, internet access was necessary to participate in the survey, potentially introducing a selection bias. The anonymous nature of the online surveys did not allow for potential motives for non-response to be investigated. We relied on self-reports, which introduced recall, sampling, and selection bias, thus reducing the representativeness and generalizability of our findings. Although we did not study a representative sample of the general population, the sociodemographic characteristics in our large study sample very likely reflect the climbing community. As an example, an older study from 2012 reported data on Austrian climbers with less than 200 participants with an average age of about 30 years [35]. To further elucidate the reasons for changed activity patterns, predominant climbing preferences, and life satisfaction among climbers during the COVID-19 pandemic, future research should employ mixed-method and longitudinal study designs.

## 6. Conclusions

The multifaceted mid- and longer-term effects of the COVID-19 pandemic pose an unprecedented public health threat. This cross-sectional study assessed the perceived impacts of the COVID-19 pandemic on health, climbing, and life satisfaction levels and respective gender and age differences among German-speaking climbers. Evidence is accumulating that outdoor fitness gained importance due to COVID-19 for several reasons, and this was also true for the surveyed climbers. During lockdowns, outdoor training was still possible in many regions and thus was an inexpensive alternative to stay mentally and physically fit during this severe health crisis. Given the higher air ventilation and space availability, training is less dangerous in regard to infection risk outside compared to indoors. Fitness studios and sports organizations should therefore consider moving equipment and activities outside whenever possible to increase safety for the users. In our study, life satisfaction of climbers decreased significantly during the pandemic, especially in females, and across all age groups. While a decrease in physical activity increases the risk of non-communicable diseases such as obesity, social isolation leads to mental impairments such as an increase in depressive symptoms and decrease in life satisfaction. For a mentally and physically healthy post-pandemic society, outdoor and indoor sports activities should be continued. In particular, strategies must be developed for the continuation of outdoor sports activities such as following hygiene measures or limiting the number of athletes in potentially crowded places and narrow areas.

## Figures and Tables

**Table 1 ijerph-20-01964-t001:** Sociodemographic characteristics of the study population (n = 1028).

	N	%
**Gender**		
Female	518	50.39
Male	507	49.32
Non-binary	3	0.29
**Age groups**		
Age low (18–32 years)	529	51.50
Age high (>32 years)	499	48.50
**Education level**		
Primary education	162	15.76
Secondary education	285	27.72
Tertiary education	581	56.52
**Country of residence**		
Austria	811	78.89
Other countries	217	21.11
**Area of residence**		
Rural area	424	41.25
Town	308	29.96
City	296	28.79
**Total**	**1028**	**100.0**

**Table 2 ijerph-20-01964-t002:** Subgroup-specific responses to items related to the COVID-19 pandemic and health.

Items	Group	Mean	SD	Cohen’s d	*p* Value
**I think the COVID-19 pandemic is dangerous.**	Total	3.82	1.14		
	Females	3.81	1.12	0.029	0.639
	Males	3.84	1.15		
	Age low	3.84	1.14	0.019	0.758
	Age high	3.81	1.13		
**Due to the COVID-19 pandemic, I am worried about my health.**	Total	2.64	1.27		
	Females	2.81	1.27	0.280	<0.001 **
	Males	2.46	1.25		
	Age low	2.64	1.28	0.009	0.879
	Age high	2.65	1.28		
**Due to the COVID-19 pandemic, I’m worried about getting infected while climbing.**	Total	2.04	1.14		
	Females	2.17	1.19	0.242	<0.001 **
	Males	1.9	1.06		
	Age low	2.11	1.17	0.123	0.049 *
	Age high	1.97	1.09		
**Due to the COVID-19 pandemic, I disinfect my hands before and after climbing.**	Total	3.19	1.46		
	Females	3.31	1.46	0.165	0.008 *
	Males	3.07	1.45		
	Age low	3.20	1.45	0.007	0.911
	Age high	3.19	1.46		

Note: Agreement with the items: 1: strongly disagree; 2: disagree; 3: neutral; 4: agree; 5: strongly agree. Effect sizes (Cohen’s d) and *p* values from independent-samples *t* tests: * <0.05, ** <0.001.

**Table 3 ijerph-20-01964-t003:** Responses to items related to climbing (single choice on most applicable climbing venue, climbing style, and outdoor sports frequency) before and during the COVID-19 pandemic.

Items	Before COVID-19		During COVID-19		Diff.%
	N	%	n	%	
**Climbing venue**					
Climbing gym	586	57.0	242	23.5	−33.50
Nature	410	39.9	642	62.5	22.60
Others	32	3.1	37	3.6	0.50
Not at all	0	0	107	10.4	10.40
**Climbing style**					
Bouldering	301	29.3	222	21.6	−7.70
Sport climbing	570	55.4	519	50.5	−4.90
Alpine climbing and others	157	15.3	287	27.9	12.60
**Outdoor sports frequency**					
0–1 h	376	36.6	461	44.8	8.20
>1–2 h	191	18.6	200	19.5	0.90
>2–5 h	241	23.4	202	19.6	−3.80
>5 h	220	21.4	165	16.1	−5.30

**Table 4 ijerph-20-01964-t004:** Responses to the Short Life Satisfaction Questionnaire-Lockdowns (SLSQL) before and during the COVID-19 pandemic.

Items	Before COVID-19		During COVID-19		Diff. Mean	-Diff. %	*t* Test	Cohen’s d	*p* Value
	Mean	SD	Mean	SD					
In most ways, my life is close to my ideal.	4.30	0.96	3.14	1.39	1.16	23.22	27.36	0.85	<0.001 **
So far, I have gotten the important things I want in life.	4.37	0.94	3.33	1.45	1.04	20.82	24.90	0.78	<0.001 **
I am satisfied with my life.	4.54	0.88	3.59	1.43	0.94	18.88	22.49	0.70	<0.001 **
Total score Life Satisfaction (total)	4.40	0.81	3.35	1.28	1.05	20.97	27.90	0.87	<0.001 **
Total score Life Satisfaction (females)	4.40	0.83	3.26	1.31	1.14	22.82	20.90	0.92	<0.001 **
Total score Life Satisfaction (males)	4.41	0.77	3.46	1.25	0.96	19.13	18.62	0.83	<0.001 **
Total score Life Satisfaction (Age low)	4.43	0.80	3.35	1.28	1.08	21.59	20.50	0.90	<0.001 **
Total score Life Satisfaction (Age high)	4.37	0.82	3.36	1.30	1.02	20.32	18.93	0.87	<0.001 **

Note: Agreement with the items: 1: strongly disagree; 2: disagree; 3: neutral; 4: agree; 5: strongly agree. ** all *p* values from paired-samples *t* tests < 0.001.

## Data Availability

The data supporting the findings of this study are available from the corresponding author upon request.

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
