# Peer review of "Sports and the Pandemic: The Impact of COVID-19 on Active Living and Life Satisfaction of Climbers"

_ijerph, 2023, doi:10.3390/ijerph20031964_

Round 1
Reviewer 1 Report
The article is generally good. However, some small aspects are suggested to be improved a little more.
Comment 1 : Is there an IRB and the consent letter signed by the student or parents?
Comment 2 : The results of t-test need report the unbiased effect size hedge'g in Table 2.
Comment 3 : 3.4 Life satisfaction change in climbers, the second paragraph in the consent used independent samples t-test alone to compare gender before COVID-19 and during COVID-19. Comparisons between gender before COVID-19 alone were not significant. In addition, the results could not control for prior effects, suggesting that repeated ANOVA could be used to analyze inter- and intra-group results better.
Comment 4 : 2.3 Statistical data analysis in the article, questionnaires should be introduced the contents in detail, meanwhile, the values of internal consistency coefficient a should be stated.
Author Response
Reviewer 1
The article is generally good. However, some small aspects are suggested to be improved a little more.
Comment 1:
Is there an IRB and the consent letter signed by the student or parents?
Authors’ response:
Dear Reviewer 1!
We thank you for the overall favorable evaluation of our research paper and the constructive feedback, which we incorporated in the revised version of the manuscript accordingly!
As for the first comment, we surveyed adults and also reported that we asked for ethical permission to conduct the survey, and all participants gave their informed consent.
Please see the methods section for this statement:
“Prior to data collection, ethical approval was granted from the institutional ethical committee of the Medical University of Vienna, Austria, on 9 October 2020. The study was conducted following the ethical standards laid down in the Declaration of Helsinki.”
Comment 2:
The results of t-test need report the unbiased effect size hedge'g in Table 2.
Authors’ response:
We agree and added effect sizes (i.e. Cohen’s d) to Table 2, hence, those are all rather small, in view of the non-significant differences for most of the items. Please note, that for very small sample sizes (n<20) Hedges’ g is chose over Cohen’s d. For sample sizes >20, the results for both statistics are roughly equivalent. So, we used Cohen’s d, as our sample size was far over n=1000, and hope, that this is ok.
Comment 3:
3.4 Life satisfaction change in climbers, the second paragraph in the consent used independent samples t-test alone to compare gender before COVID-19 and during COVID-19. Comparisons between gender before COVID-19 alone were not significant. In addition, the results could not control for prior effects, suggesting that repeated ANOVA could be used to analyze inter- and intra-group results better.
Authors’ response:
We thank you for this suggestion. As for life satisfaction, we used the standardized Short Life Satisfaction Questionnaire for Lockdowns (SLSQL) before and during the crisis and employed the statistical measures (i.e. dependent sample t-test) published with this scale by several authors, which we cited. The most important result refers to the change in life satisfaction. Independently and not related to these publications, we compared the mean values for the Life satisfaction score of the subgroups age and gender to answer the question whether the ratings in those groups differ statistically significantly, which is normally done with t tests, among other methods that would be able to answer the question statistically.
Comment 4:
2.3 Statistical data analysis in the article, questionnaires should be introduced the contents in detail, meanwhile, the values of internal consistency coefficient a should be stated.
Authors’ response:
Thanks for this comment, we modified this section. Notably, we measured internal consistency using Cronbach’s alpha, and this is already reported in the results section. In response to this comment, we just moved them to the end of the methods section.
“Internal consistency (Cronbach’s alpha) for the three SLSQL items before (i. e. 0.841) and for the three SLSQL items during COVID-19 (i.e. 0.889) were high.”
Again, thanks a lot for your time and effort!
Reviewer 2 Report
Dear authors,
The submitted article has a number of shortcomings that need to be improved in order to be considered for publication.
- The questionnaire has not been validated. Furthermore, the relevant reliability and internal consistency tests have not been carried out to ensure its validity with the sample used.
- Sampling is voluntary, which considerably reduces the possibility of being able to generalise conclusions.
- The analyses are not properly designed, since in order to be able to use a Student's t-test, first of all, they should have checked the normality of the sample and the homogeneity of its variance. This is highly unlikely given the nature of the sampling.
- Many of the variables analysed are categorical, so it is not feasible to use a Student's t-test. Instead, tests for categorical variables such as Chi-square should be used.
I hope these comments will help you to improve your work.
Author Response
Reviewer 2
Dear authors,
The submitted article has a number of shortcomings that need to be improved in order to be considered for publication.
Comment:
- The questionnaire has not been validated. Furthermore, the relevant reliability and internal consistency tests have not been carried out to ensure its validity with the sample used.
Authors’ response:
Dear Reviewer 2!
We thank you for the evaluation of our research paper and the constructive feedback, which we incorporated in the revised version of the manuscript accordingly!
We agree that a reliable survey tool is important, so we would like to highlight that we performed a state-of-the art survey using an already published questionnaire in German language including the standardized Short Life Satisfaction Questionnaire for Lockdowns (SLSQL), (see citations) which we pretested. As for measuring Internal Consistency, we used Cronbach’s alpha, the most common measure in this respect, and reported the measures, which notably, indicated a high internal consistency, as stated here:
“Internal consistency (Cronbach’s alpha) for the three SLSQL items before (i. e. 0.841) and for the three SLSQL items during COVID-19 (i.e. 0.889) were high.”
Comment:
- Sampling is voluntary, which considerably reduces the possibility of being able to generalise conclusions.
Authors’ response:
We totally agree, as we are aware of the benefits, but also the shortcomings of an online survey, so we raised this point, among others, in the limitations section.
“We relied on self-reports, which introduced recall bias, sampling and selection bias, thus reducing the representativeness and generalizability of our findings”.
Comment:
- The analyses are not properly designed, since in order to be able to use a Student's t-test, first of all, they should have checked the normality of the sample and the homogeneity of its variance. This is highly unlikely given the nature of the sampling.
- Many of the variables analysed are categorical, so it is not feasible to use a Student's t-test. Instead, tests for categorical variables such as Chi-square should be used.
I hope these comments will help you to improve your work.
Authors’ response:
We very carefully planned and designed the study, with a notably large sample size of n over 1000, in accordance with the already published methods and international literature (please see the according references, e.g. Ammar et al. 2020 with over 330 citations by now). To increase comparability of the methods with the pertinent literature, we used the methods and statistical procedures used therein, as explained in the methods section, and we did not use alternative and equivalent methods, just to achieve the same outcomes. We are well aware of the long-standing scholar discussion of how to analyze categorical variables from surveys. Nowadays, the majority of authors use t test because it's simpler conceptually and straightforward to compute a measure of effect size. So, the statistical procedures were chosen after statistical consulting for a clear reason based on the most relevant contributions in the field. We hope this is ok with you.
Again, we thanks you for your valuable feedback!
Reviewer 3 Report
This manuscript is an original investigation and the topic under study is relevant. From the point of view of methological rigor, the problem of the study was identified, referring its objetives,and presented some scientific evidence on the subject of the study. The research variables are defined and the procedures and instruments for the research were explained. The results are presented, being complemented with a discussion. Reflections on the limitations of the study were carried out. The conclusions are enlightening and the references are adjusted to the topic under discussion. The text presents, in terms of writing, several marks of orality that are suggested to be reviewed. Below there are some suggestions and questions for you to consider:
Line 12: “Thus, were surveyed…”
Line 20: “…in this study sample…”
Line 37: “…is an useful…”
Lines 54 and 55: “The main goal of this article is to study the impact of COVID-19…”
Line 89: “No incentives were offered. Were not used randomized…”
Line 105: “The website administrators of the most prominent climbing organizations were contacted, to send the questionnaire link via…”
Line 130 to 137: when the subjects answered the survey they were in a COVID-19 context, don´t you think that this can influence the answers they gave about their satisfaction before COVID-19? How did you control this issue? I suggest that you describe I you control this question.
Line 140: “It was used…”
Line 141: “ It was performed…”
Line 143: “It was set statistical significance at… It was used paired
Lines 146, 147and 149 : “It was conducted… It was calculated… It was interpreted…”
Line 155: “It were excluded…”
Line 173: “It was found…”
Lines224, 225 and 230: It is suggested to remove this statements, since this study did not aim to carry out a systematic review on the subject.
Line 241: “…in this sample of climbers”
Line 245: “In contrast, this study…”
Line 294: “In contrast, this study…”
Line 319: “…it is assume…”
Line 324: “It was assessed…” “So, this study…”
Line 375: “It was compared…”
Line 379: “It was collected…”
Author Response
Reviewer 3
This manuscript is an original investigation and the topic under study is relevant. From the point of view of methological rigor, the problem of the study was identified, referring its objetives,and presented some scientific evidence on the subject of the study. The research variables are defined and the procedures and instruments for the research were explained. The results are presented, being complemented with a discussion. Reflections on the limitations of the study were carried out. The conclusions are enlightening and the references are adjusted to the topic under discussion. The text presents, in terms of writing, several marks of orality that are suggested to be reviewed. Below there are some suggestions and questions for you to consider:
Line 12: “Thus, were surveyed…”
Line 20: “…in this study sample…”
Line 37: “…is an useful…”
Lines 54 and 55: “The main goal of this article is to study the impact of COVID-19…”
Line 89: “No incentives were offered. Were not used randomized…”
Line 105: “The website administrators of the most prominent climbing organizations were contacted, to send the questionnaire link via…”
Line 130 to 137: when the subjects answered the survey they were in a COVID-19 context, don´t you think that this can influence the answers they gave about their satisfaction before COVID-19? How did you control this issue? I suggest that you describe I you control this question.
Line 140: “It was used…”
Line 141: “ It was performed…”
Line 143: “It was set statistical significance at… It was used paired
Lines 146, 147and 149 : “It was conducted… It was calculated… It was interpreted…”
Line 155: “It were excluded…”
Line 173: “It was found…”
Lines224, 225 and 230: It is suggested to remove this statements, since this study did not aim to carry out a systematic review on the subject.
Line 241: “…in this sample of climbers”
Line 245: “In contrast, this study…”
Line 294: “In contrast, this study…”
Line 319: “…it is assume…”
Line 324: “It was assessed…” “So, this study…”
Line 375: “It was compared…”
Line 379: “It was collected…”
Authors´ response:
Dear Reviewer,
we thank you for the overall favorable evaluation of our research paper and the constructive feedback including some typos, which we corrected in the revised version of the manuscript!
We used active voice whenever possible to increase readability of the text. In response to your comments, we changed those expressions to passive voice.
As for the COVID-10 context, as done in the cited literature with similar methods and especially with self-reports, it is crucial to set a primer for the time period of interest, so we defined what we mean with before and during the crisis. In response to this comment and to increase clarity, we modified the according phrase in the methods section:
“An explanatory text defined that “before COVID-19” means the time period before 16 March 2020, and “during COVID-19” means the time period after this date.”
As for the discussion, we modified the first part according to your comments:
“The present study examined the effects of the COVID-19 pandemic on active living and life satisfaction in German-speaking climbers. This unprecedented health crisis fundamentally changed everyday life and impacted people´s former patterns of recreational activities, eating behavior, and social interactions. As a result, measures of health and wellbeing decreased substantially in response to the crisis.”
Again, thanks a lot for your time and effort!
Reviewer 4 Report
I find the study interesting and relevant. Some comments The introduction is good but you should look at the sentence 48. something is lacking. At the end of the introduction, did you have any hypoteses for the study.? What do you expect to find out. This because you have chosen to test the results statistically. (Sentence 80- 81) The study design and the measures are well described. Chapter 3.1 Study population should be moved to cahpter 2 study design. Everything that you descibe here is also in the table 1 which you havent refered to. You can delete some of the information in the text. There is no chapter 3.2 and 3.3 only 3.1 and 3.4.
The information in line 185 and 186 is not necessary (table 2). The only thing you have to include is the sign. level. The rest is described earlier. This is the same for table 4 line 220 and 221.
Discussion
What do you define as individual climbers? line 326.
When you have come up with some hypoteses you can come up with some cunclusions as well. The discussion is over all ok and relevant
Limitations are ok
You could also include in the conclusion that use of liquid chalk which contains between 40 -80 % alcohol will decrease the risk of infection.
Author Response
Reviewer 4
I find the study interesting and relevant. Some comments The introduction is good but you should look at the sentence 48. something is lacking.
Authors’ response:
Dear Reviewer,
we thank you for the overall favorable evaluation of our research paper and the constructive feedback, which we incorporated in the revised version of the manuscript accordingly!
We added the missing word in phrase 48:
“Also, it is not possible to practice all sports in the home environment….”
Comment:
At the end of the introduction, did you have any hypoteses for the study.? What do you expect to find out. This because you have chosen to test the results statistically. (Sentence 80- 81)
Authors’ response:
In response to this comment und to increase clarity, we modified the text and added the following to the end of the introduction section:
“So far, knowledge on the influence of the pandemic on climbing, a popular mass sport with different indoor- and outdoor-based styles, is scarce. Based on the pertinent literature negative effects of the crisis on physical activity and mental health, we hypothesized that climbers in German-speaking countries were also affected by lockdowns, curfews, and gym closures. We were also interested in elucidating whether socio-demographic variables such as gender and age impacted perceived impacts of the pandemic on health, climbing, and life satisfaction levels. Thus, the current study examines whether climbing frequency and preferences of indoor or outdoor climbing facilities as well as levels of life satisfaction changed during the COVID-19 pandemic. Using a cross-sectional online survey including the standardized Short Life Satisfaction Questionnaire for Lockdowns (SLSQL), we surveyed a nonrepresentative convenience sample of German-speaking climbers and compared ratings before vs. during the COVID-19 crisis in the climbing context.”
Comment:
The study design and the measures are well described. Chapter 3.1 Study population should be moved to cahpter 2 study design. Everything that you descibe here is also in the table 1 which you havent refered to. You can delete some of the information in the text. There is no chapter 3.2 and 3.3 only 3.1 and 3.4.
The information in line 185 and 186 is not necessary (table 2). The only thing you have to include is the sign. level. The rest is described earlier. This is the same for table 4 line 220 and 221.
Authors’ response:
We thank you for this remark. We agree and moved the according parts and renumbered the subsections. Notably, table 1 was referred to in this phrase “The final sample included 1028 participants, with nearly half females (n= 518, 50.39%) and males (n= 507, 49.32%), and three diverse people (0.29%, Table 1).” Also, we removed the unnecessary text from the table legends.
Comment:
Discussion
What do you define as individual climbers? line 326.
Authors’ response:
In response to this comment, we modified this phrase to avoid misunderstanding and increase clarity to the following:
“We assessed the change in life satisfaction levels before vs. during the pandemic in the general sample and between subgroups, acknowledging that, among other factors, age and gender influence this score.”
Comment:
When you have come up with some hypoteses you can come up with some cunclusions as well. The discussion is over all ok and relevant
Limitations are ok
You could also include in the conclusion that use of liquid chalk which contains between 40 -80 % alcohol will decrease the risk of infection.
Authors´ response:
We agree. In response to these comments, we modified the conclusion, also adding the following:
“This cross-sectional study assessed the perceived impacts of the COVID-19 pandemic on health, climbing, and life satisfaction levels and respective gender and age differences among German-speaking climbers.”
Also, we added some thoughts on liquid chalk to the discussion section, as we find this aspect really important when it comes to hygiene recommendations:
“Liquid chalk, which is a type of climbing chalk that is made with a mixture of powdered chalk and alcohol, has become increasingly popular among climbers. The alcohol in liquid chalk is used as a carrier to help keep the chalk on the hands, and can be found in varying concentrations (from 40-80%). It is important to note that liquid chalk is primarily used to dry and grip the hands, not to decrease the risk of infection. Consequently, hygiene recommendations in times of a virus pandemic should include disinfecting and washing hands before and after climbing, and cleaning surfaces such as climbing holds frequently and regularly.”
Again, thanks a lot for your time and effort!
Round 2
Reviewer 2 Report
Dear authors,
After reviewing the requested changes, I consider the manuscript suitable for publication.
Kind regards